# AUTO-ENCODING INVERSE REINFORCEMENT LEARNING

## ABSTRACT

Reinforcement learning (RL) provides a powerful framework for decision-making, but its application in practice often requires a carefully designed reward function. Inverse Reinforcement Learning (IRL) has shed light on automatic reward acquisition, but it is still difficult to apply IRL to solve real-world tasks. In this work, we propose Auto-Encoding Inverse Reinforcement Learning (AEIRL), a robust and scalable IRL framework, which belongs to the adversarial imitation learning class. To recover reward functions from expert demonstrations, AEIRL utilizes the reconstruction error of an auto-encoder as the learning signal, which provides more information for optimizing policies, compared to the binary logistic loss. Subsequently, we use the derived objective functions to train the reward function and the RL agent. Experiments show that AEIRL performs superior in comparison with state-of-the-art methods in the MuJoCo environments. More importantly, in more realistic settings, AEIRL shows much better robustness when the expert demonstrations are noisy. Specifically, our method achieves $16.1\%$ relative improvement compared to the best baseline FAIRL on clean expert data and $46.5\%$ relative improvement compared to the best baseline PWIL on noisy expert data both with the metric overall averaged scaled rewards.

## 1 INTRODUCTION

Reinforcement learning (RL) provides a powerful framework for automating decision making. However, RL still requires significantly engineered reward functions for good practical performance. To make RL more applicable in the real-world, it is important to learn a reward function from expert demonstrations. Imitation learning offers the instruments to learn policies directly from the data, without an explicit reward function. Imitation learning enables the agents to learn to solve tasks from expert demonstrations, such as helicopter control (Abbeel et al., 2006; 2007; Ng et al., 2004; Coates et al., 2008; Abbeel et al., 2008a; 2010), robot navigation (Ratliff et al., 2006; Abbeel et al., 2008b; Ziebart et al., 2008; 2010), and building controls (Barrett & Linder, 2015).

The goal of imitation learning is to extract the expert policies from the expert demonstrations without the access to the reward signal from the environment. The algorithms in this field can be divided into two broad categories: behavioral cloning (BC) and Inverse Reinforcement Learning (IRL). BC formulates the learning task as a supervised learning problem, which learns a mapping from the states to the actions, using the expert trajectories. However, BC methods suffer from the problem of compounding errors, i.e., covariate shift, which only learn the actions of the expert but not reason about what the expert attempts to achieve. In the contrast, IRL directly infers the underlying reward function from the data, and then teaches the RL agent to learn a policy to achieve the highest accumulated reward. Empirical results show that IRL methods are more efficient than BC methods in multi-step decision making tasks. However, how to make IRL stable and efficient to use is still subject to research.

Current IRL methods, such as adversarial imitation learning approaches, model the reward function as a discriminator to learn the mapping from the state-action pair to a scalar value, i.e., the reward (Ho & Ermon, 2016; Fu et al., 2017; Ghasemipour et al., 2020). However, this formulation may be easily over-confident to focus on the minor differences between the state-action features of the expert and generated samples. To the best of our knowledge, this is the first instance of formulating the reward function as an auto-encoder, which has the ability to learn full scale differences between

expert and alternative policies. The derived formulation ensures the reward signal is informative, which is efficient for optimization. Additionally, the encoding-decoding process empowers the denoising capability and makes the agent robust to the noisy expert, in more realistic settings. In the experiments, we show that the proposed method AEIRL achieves the best overall performance on both clean and noisy expert demonstrations.

Our contributions are three-fold:

- We propose the Auto-Encoding Inverse Reinforcement Learning (AEIRL) architecture, which models the reward function as a surrogate function using the reconstruction error of an auto-encoder. The reconstruction error provides more informative learning signals, compared to the binary logistic loss.

- To show the contributing factors of our method, we conduct ablation studies based on different distribution divergences and alternative auto-encoders. Experiments show that they achieve comparable results which indicates the encoding-decoding process is the major contributing factor for our method.

- The experimental results on the MoJoCo tasks show that our method outperforms state-of-the-art imitation learning methods on both clean and noisy expert demonstrations. Empirical analysis show that our learned reward function can be more informative and robust. Furthermore, the learning processes of our methods are also more stable in general.

## 2 RELATED WORK

### 2.1 INVERSE REINFORCEMENT LEARNING

Adversarial imitation learning such as GAIL, AIRL, DAC, $f$-GAIL, EAIRL, FAIRL (Ho & Ermon, 2016; Fu et al., 2017; Kostrikov et al., 2019; Zhang et al., 2020; Qureshi et al., 2019; Ghasemipour et al., 2020) formulates the learned reward function as a discriminator that learns to differentiate expert transitions from non-expert ones. Among these methods, GAIL (Ho & Ermon, 2016) considers the Jensen-Shannon divergence, while AIRL (Fu et al., 2017) considers the Kullback-Leibler (KL) divergence. DAC (Kostrikov et al., 2019) extends GAIL to the off-policy setting and significantly improves the sample-efficiency of adversarial imitation learning. Furthermore, $f$-divergence is utilized in $f$-GAIL (Zhang et al., 2020), which is considered more sample-efficient. Recently, FAIRL utilizes the forward KL divergence (Ghasemipour et al., 2020) and achieves better performance than AIRL (Fu et al., 2017) comprehensively, but it is still not robust enough. However, these methods rely heavily on a carefully tuned discriminator, which might easily overfit to the minor differences between the expert and the generated samples. In comparison, our auto-encoder based reward function helps to learn the full scale differences between the expert and generated samples, which provides more informative reward signals.

The robustness of adversarial imitation learning is also questionable with imperfection in observations (Stadie et al., 2017; Berseth & Pal, 2020), actions, transition models (Gangwani & Peng, 2020; Christiano et al., 2016), expert demonstrations (Brown et al., 2019; Shiarlis et al., 2016; Jing et al., 2020) and their combinations (Kim et al., 2020). Previous robust IRL methods require the demonstrations to be annotated with confidence scores (Wu et al., 2019; Brown et al., 2019; Grollman & Billard, 2012), when the expert data is noisy. However, these annotations are rather expensive. Compared to this, our auto-encoder based reward function helps to denoise the expert data through the encoding-decoding process. Our method AEIRL is relatively succinct and robust to noisy expert demonstrations and does not require any annotated data.

Another category of IRL uses an offline similarity function to estimate the rewards (Boularias et al., 2011; Klein et al., 2013; Piot et al., 2016). The idea of these methods is still inducing the expert policy by minimizing the distance between the state action distributions of the expert and sampled trajectories. To the best of our knowledge, the most powerful method in this category is Primal Wasserstein Imitation Learning (PWIL) (Dadashi et al., 2021), which utilizes the upper bound of its primal form as the optimization objective. The advantage of these methods is that they are relatively more robust compared to adversarial imitation learning methods, when the expert data is noisy. However, the performance of these methods heavily depends on the similarity measurement, and

therefore, it varies greatly on different tasks. Compared to PWIL, our method achieves superior performance.

## 2.2 AUTO-ENCODING BASED GANS

Auto-encoders have been successfully applied to improve the training stability and modes capturing in GANs. Auto-encoding based GANs can be classified into three categories: (1) utilizing an auto-encoder as the discriminator such as energy-based GANs (Zhao et al., 2016) and boundary-equilibrium GANs (Berthelot et al., 2017); (2) using a denoising auto-encoder to derive an auxiliary loss for the generator (Warde-Farley & Bengio, 2017); (3) combining variational auto-encoder and GANs to generate both vivid and diverse samples by balancing the objective of reconstructing the training data and confusing the discriminator (Larsen et al., 2016). Our method AEIRL takes inspirations from EBGAN (Zhao et al., 2016) and utilizes an auto-encoder as the reward function and derives the efficient reconstruction error based surrogate reward signal and its corresponding objective functions.

## 3 BACKGROUND

A Markov decision process (MDP) is a tuple $(S, A, T, \gamma, P, r)$. In this tuple, $S$ is a state space; $A$ is an action space; $T$ is a probability matrix for state transitions; $\gamma \in (0, 1]$ is a discount factor; $P$ is a initial-state transition distribution; and $r : S \times A \to \mathbb{R}$ is a reward function. Additionally, we also define a stochastic policy $\pi$, which is a mapping from states to probability distributions over actions.

IRL infers the reward function using the expert demonstrations, which are assumed to be the observations of optimal behaviors (Ng & Russell, 2000). In general, IRL is formulated as a bi-level optimization process meaning that iteratively training the reward function and optimizing the policy. Assume that we are given an expert policy $\pi_E$, IRL (Ziebart et al., 2008; 2010) fits a reward function from a family of functions $\mathcal{R}$ with the optimization problem:

$$\min_{r \in \mathcal{R}} \left( \max_{\pi \in \Pi} \mathbb{E}_\pi[r(s, a)] \right) - \mathbb{E}_{\pi_E}[r(s, a)] \tag{1}$$

Moreover, the expert policy $\pi_E$ will only be provided by a set of expert demonstrations, so the expected reward of $\pi_E$ is estimated by these trajectories.

IRL looks for a reward function $r(s, a)$ that assigns high values to the expert policy and low values to other policies. Therefore, it allows the expert policy to be found with a certain reinforcement learning procedure:

$$\mathrm{RL}(r) = \arg \max_{\pi \in \Pi} \mathbb{E}_\pi[r(s, a)] \tag{2}$$

The RL process will induce the expert policy via maximizing the expected cumulative rewards. Meanwhile the entropy term can be optionally added to the reinforcement learning objective to encourage the exploration in policy searching. Typically, IRL models the reward function as a discriminator which can be easily overfit to the expert data. Also its training stability is questionable once confronted with even a little noise in the expert data. Therefore, we propose auto-encoding IRL to utilize an auto-encoder based reward function which achieve strong performance both on clean and noisy expert demonstrations.

## 4 AUTO-ENCODING INVERSE REINFORCEMENT LEARNING

### 4.1 OVERVIEW

The reward function in GAIL is a discriminator, which attempts to assign high values to the regions near the expert demonstrations and assign low values to the other regions. However, this form of reward function could be easily overfitting to the expert data. Consider the CartPole balancing task, the state of the environment is a feature consisting of position, angle, angle's rate, and cart velocity, while the action is moving left or right. Here, we assume that all the expert states' velocity is 2 for example. When the generated states' velocity is 1.9 and other dimensions of state action pairs are

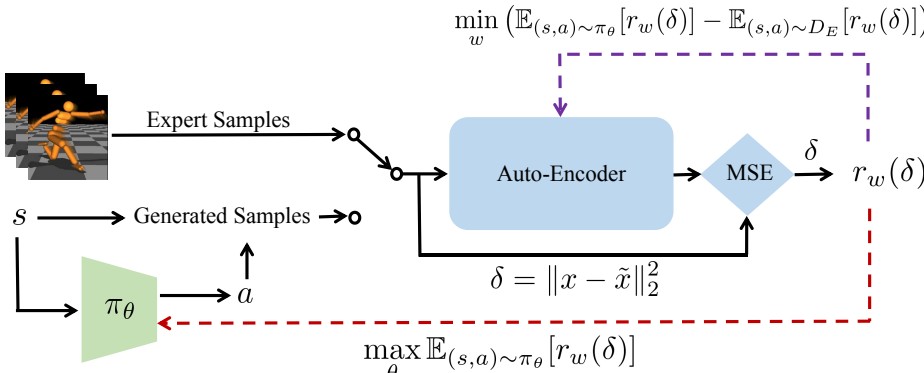

$$\min_w \left( \mathbb{E}_{(s,a)\sim\pi_\theta}[r_w(\delta)] - \mathbb{E}_{(s,a)\sim D_E}[r_w(\delta)] \right)$$

Figure 1: Adversarial training framework of auto-encoding inverse reinforcement learning. The auto-encoder computes the reconstruction error for these two mini-batches of data examples and optimize the objectives. The surrogate reward function provides the signal to the agent.

the same as the expert's, the discriminator of GAIL would still give a low reward on these generated state action pairs. However it may actually perform very well on the goal of mimicking the expert's behaviors. In other words, the reward function in GAIL on this example could easily overfit to the minor differences between the expert and the sampled data, while missing the underlying goal of the expert.

In our paper, we propose an auto-encoder based reward function for inverse reinforcement learning. It utilizes the reconstruction error of the state action pairs to yield an informative reward signal. The reconstruction error based reward signal significantly retains the information of state action pairs, rather than focusing on the minor differences. Such a reward signal wouldn't lead to overconfidence in distinguishing the expert and the generated samples. Recall the CartPole balancing example, the mean square error between states' velocity 1.9 and 2 is very small. And it could still feed a good reward to the agent under this situation. Thus, the reconstruction error based reward signal focuses on the full scale differences between the expert and generated state action pairs rather than the minor parts.

This yields a much more informative reward signal. Figure 5 (See in Section 5.4), shows a more informative reward signal recovered with our method. Furthermore, the training process of the policy becomes smoother in general, which is shown in Figure 11 (See in Appendix A.6).

The expert demonstrations usually contains noise bacause we sample these human trajectories with sensors and other devices in the real world. Therefore, adversarial imitation learning, such as GAIL, might be easily affected to learn the noisy expert behaviors, which are not the real intentions of the expert. The reward function under this situation could overfit to the noisy features. Recall the CartPole balancing example, we assume that the expert states' velocity is 2 but it tends to be $2+\delta$ due to the noisy sampling process. When the learned policy is good enough as the sampled states' velocity, the discriminator in GAIL still takes it as a bad policy, and therefore, it loses its efficacy, when learning from noisy demonstrations.

Figure 2 depicts the denoising process of an auto-encoder. When the auto-encoder is trained to minimize the averaged squared errors, the vector points approximately towards the nearest points on the manifold, since the auto-encoder estimates the center of mass of the clean points (Goodfellow et al., 2016). Therefore, the reconstruction error of the auto-encoder can eliminate some effects when learning from noisy expert demonstrations and

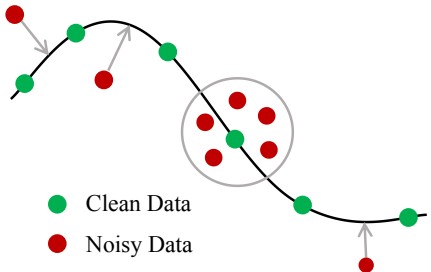

Figure 2: The clean data points lie near a low dimensional manifold illustrated with the bold black line. The gray circle shows the noisy sampling process. And the auto-encoder learns to denoise the noisy data to clean points.

make the reward signal more robust. Recall the CartPole example, the reconstruction error of the auto-encoder is $\delta^2$ which doesn't play an important role in the rewards for the agent.

## 4.2 METHOD

Our approach is to minimize the distance between the state action distribution of the policy $\pi_\theta$ and the expert demonstrations $D_E$. The distance we used in our method is the Wasserstein distance (Arjovsky et al., 2017):

$$d(\pi_E, \pi_\theta) = \sup_{r_w} \mathbb{E}_{\pi_E}[r_w(s, a)] - \mathbb{E}_{\pi_\theta}[r_w(s, a)], \tag{3}$$

where the reward function network's parameters are denoted as $w$ and the policy network's parameters are represented as $\theta$. We choose the metric functions in Wasserstein distance as a class of reward functions from neural network functions for computational convenience. Minimizing this distance is actually inducing the expert policy from expert demonstrations. Therefore, the optimization of the policy $\pi_\theta$ and recovering period of the reward function $r_w(s, a)$ forms a bi-level optimization problem, which can be formally defined as:

$$\min_{\pi_\theta} \max_{r_w} \mathbb{E}_{(s,a)\sim D_E}[r_w(s, a)] - \mathbb{E}_{(s,a)\sim \pi_\theta}[r_w(s, a)]. \tag{4}$$

This leads to an adversarial formulation for inverse RL. The outer level minimization with respect to the policy leads to a learned policy which is close to the expert. The inner level maximization recovers a reward function which attributes higher values to regions close to the expert data, and penalizes all other regions.

In traditional adversarial imitation learning methods, such as WGAIL (Arjovsky et al., 2017; Ho & Ermon, 2016) (Details See in Appendix A.2.3), the reward function is defined as

$$r_w(s, a) = D_w(s, a), \tag{5}$$

where $D_w(s, a)$ is the output of the discriminator.

In our method, we use an auto-encoder based surrogate reward function, which is defined as:

$$r_w(s, a) = 1/(1 + \text{AE}_w(s, a)), \tag{6}$$

where AE is the reconstruction error of an auto-encoder:

$$\text{AE}(x) = \|\text{Dec} \circ \text{Enc}(x) - x\|_2^2 \tag{7}$$

Here, $x$ represents the state-action pairs. It is a mean square error between the sampled state action pairs and the reconstructed samples. This form of the reward signal uses the reconstruction error of an auto-encoder to score the state action pairs in trajectories. It is a monotonically decreasing function over the reconstruction error of the auto-encoder. Low reconstruction errors ensure high reward values for state action pairs and vice versa. Section 4.1 tells that this form of reward signal focuses more on the full scale differences between the expert and generated samples and can help to denoise the expert demonstrations.

Training the auto-encoder is an adversarial process considering the objective 4, which is minimizing the reconstruction error for the expert samples and meanwhile maximizing this error for generated samples. So the auto-encoder based reward function training objective is to minimize:

$$\begin{aligned}
\mathcal{L} &= \mathbb{E}_{(s,a)\sim \pi_\theta}[r_w(s, a)] - \mathbb{E}_{(s,a)\sim D_E}[r_w(s, a)] \\
&= \mathbb{E}_{(s,a)\sim \pi_\theta}[1/(1 + \text{AE}_w(s, a))] - \mathbb{E}_{(s,a)\sim D_E}[1/(1 + \text{AE}_w(s, a))].
\end{aligned} \tag{8}$$

The auto-encoder learns to maximize the full scale differences between the expert and the generated samples with the adversarial objective. It leads to a better feedback signal to the agent. Furthermore, the auto-encoder also retains more information and helps to denoise the expert data via the encoding-decoding process. This can further help us to match the distribution between the expert and generated behaviors in a latent space.

The variational auto-encoder (Kingma & Welling, 2014) is also an alternative form of reward function. When we use an variational auto-encoder based reward function, the objective function for training the variational auto-encoder in our method is:

$$\begin{aligned}
\mathcal{L} =& \mathbb{E}_{(s,a)\sim D_E}[r_w(s, a) + D_{\text{KL}}(p(z|(s, a)), p_{\text{model}}(z))] \\
&- \mathbb{E}_{(s,a)\sim \pi_\theta}[r_w(s, a) + D_{\text{KL}}(p(z|(s, a)), p_{\text{model}}(z))],
\end{aligned} \tag{9}$$

where $p_{\text{model}}(z))$ is a fixed Gaussian distribution. When minimizing this objective, the latent distribution for the expert samples is widely distributed to induce more diverse expert policies.

Figure 1 depicts the architecture for AEIRL. The auto-encoder based reward function takes either expert or the generated state action pairs, and estimates the reconstruction error based rewards accordingly. The auto-encoder and the policy is iteratively optimized under this adversarial training paradigm. Algorithm 1 in Appendix A.1 depicts the pseudo code for training AEIRL.

Not limited to the utilized Wasserstein distance, other distribution divergences are also applicable to form the reward functions without loss of generality. Appendix A.5 shows other two forms of objectives and reward functions following Jensen-Shannon divergence. Experiments show that different forms of objectives achieve comparable results on almost all locomotion tasks. It indicates that an auto-encoder based reward function is the major contributing factor to improve the performance on mimicking the expert rather than the different distribution divergences and the corresponding forms of surrogate reward function.

## 5 EXPERIMENTS

We conduct the experiments on six locomotion tasks with varying dynamics and difficulty: Swimmer-v2 (Schulman et al., 2015), Hopper-v2 (Levine & Koltun, 2013), Walker2d-v2 (Schulman et al., 2015), HalfCheetah-v2 (Heess et al., 2015), Ant-v2 (Schulman et al., 2016), and Humanoid-v2 (Tassa et al., 2012). The goal for all these tasks is to move forward as quickly as possible.

The baselines we choose for comparison includes: behavior cloning (BC) (Dhariwal et al., 2017), GAIL (Ho & Ermon, 2016), WGAIL (Arjovsky et al., 2017; Ho & Ermon, 2016), FAIRL (Ghasemipour et al., 2020), and PWIL (Dadashi et al., 2021). For a fair comparison, we use TRPO (Schulman et al., 2015) as the policy search method for all the algorithms, which is implemented by OpenAI Baselines (Dhariwal et al., 2017). Experimental details, including the implementation and more results, are shown in Appendix A.2 to A.7.

### 5.1 LEARNING FROM EXPERT DEMONSTRATIONS

| Task | Walker | Hopper | Swimmer | HalfCheetah | Ant | Humanoid |
|------|--------|--------|---------|-------------|-----|----------|
| Random | $-0.6 \pm 0.3$ | $25.5 \pm 4.2$ | $-5.3 \pm 13.9$ | $-694 \pm 108$ | $-401.4 \pm 233.4$ | $110 \pm 4.1$ |
| Expert | 4904.2 | 2719.8 | 142.2 | 1969.2 | 2646.1 | 5187.9 |
| BC | $459.7 \pm 374.8$ | $396.5 \pm 435.7$ | $78.3 \pm 39.6$ | $945.2 \pm 160.7$ | $807.8 \pm 1843.9$ | $237.8 \pm 621.4$ |
| GAIL | $3023.1 \pm 254.4$ | $2487.3 \pm 80.3$ | $133.2 \pm 10.7$ | $1610.2 \pm 25.3$ | $2262.6 \pm 280.7$ | $2078.9 \pm 433.8$ |
| WGAIL | $3308.5 \pm 361.4$ | $2570.1 \pm 122.1$ | $133.4 \pm 5.7$ | $1658.4 \pm 42.7$ | $2356.5 \pm 299.2$ | $1669.6 \pm 296.3$ |
| FAIRL | $3214.0 \pm 290.6$ | $2650.4 \pm 101.8$ | $136.7 \pm 13.7$ | $1563.1 \pm 39.8$ | $2563.0 \pm 284.6$ | $1513.2 \pm 462.7$ |
| PWIL | $3511.4 \pm 130.3$ | $2809.9 \pm 47.2$ | $47.9 \pm 4.8$ | $1372.5 \pm 71.3$ | $1067.9 \pm 462.3$ | $3157.2 \pm 507.8$ |
| **Ours** | $3897.8 \pm 177.8$ | $2706 \pm 219.7$ | $\mathbf{141.5 \pm 3.7}$ | $\mathbf{1674.1 \pm 33.6}$ | $\mathbf{2715.7 \pm 93.1}$ | $\mathbf{3892.3 \pm 522.9}$ |
| **Ours-VAE** | $\mathbf{4013.2 \pm 248.7}$ | $\mathbf{2850.5 \pm 150.7}$ | $135.9 \pm 5.5$ | $1643.9 \pm 10.8$ | $2332.5 \pm 938.3$ | $2486.7 \pm 1418.9$ |

Table 1: Learned policy performance for different imitation learning algorithms on **non-noisy** expert data, evaluated by using the mode of sampling from the policy distributions. The best results are marked as **red**.

To evaluate the performance of our proposed methods and other baselines, we run the experiments on the clean expert demonstrations. The metric we used is the ground truth rewards for different learned policies. Higher rewards indicate better mimicking expert behaviors.

Figure 3 depicts the training curves of ground truth rewards for different algorithms. The training curve shows the imitation performance learning from non-noisy expert data. Table 1 shows the final learned policy performance on MuJoCo tasks with ground truth episode return.

GAIL, WGAIL and FAIRL have comparable results, while WGAIL and FAIRL are little better than GAIL comprehensively on these tasks but still not robust enough. PWIL achieves higher performance compared with GAIL, WGAIL and FAIRL on Walker2d, Hopper and Humanoid tasks but performs worse than BC on Swimmer. Thus, the performance of PWIL varies greatly on different tasks.

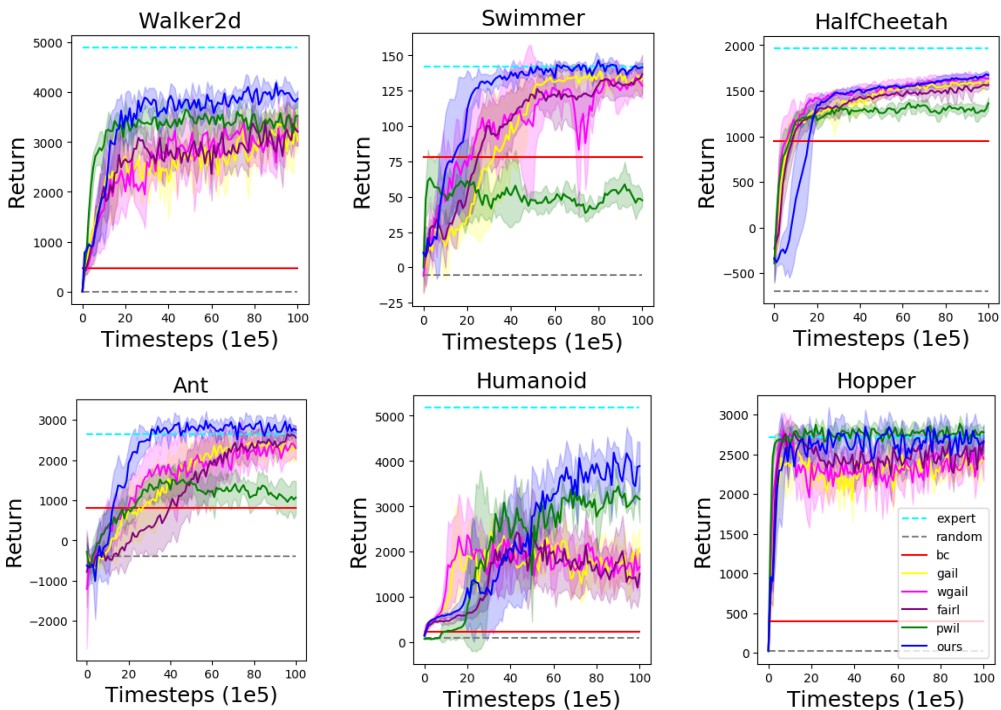

Figure 3: Mean and standard deviation return of the evaluation policy over 10 rollouts and 5 seeds, learning from **non-noisy** expert demonstrations, reported every 100k timesteps. The return is in term of the environment's ground truth reward.

| Task | Walker2d | Hopper | Swimmer | HalfCheetah | Ant | Humanoid |
|---|---|---|---|---|---|---|
| BC | $728.2 \pm 835.3$ | $195.8 \pm 15.7$ | $35.6 \pm 2.3$ | $898.1 \pm 827.9$ | $623.5 \pm 1364.1$ | $186.7 \pm 5243.6$ |
| GAIL | $1254.1 \pm 231.8$ | $1478.0 \pm 405.9$ | $\mathbf{106.4 \pm 19.7}$ | $862.3 \pm 94.2$ | $1538.3 \pm 842.9$ | $1085.0 \pm 622.8$ |
| WGAIL | $940.9 \pm 383.8$ | $2570.1 \pm 122.1$ | $95.9 \pm 15.4$ | $696.4 \pm 211.7$ | $1457.7 \pm 359.8$ | $1479.1 \pm 539.8$ |
| FAIRL | $1302.3 \pm 554.1$ | $1732.3 \pm 328.2$ | $106.2 \pm 12.3$ | $940.1 \pm 98.1$ | $-767.3 \pm 518.2$ | $1668.7 \pm 511.2$ |
| PWIL | $3052.4 \pm 346.6$ | $\mathbf{2588.9 \pm 159.7}$ | $57.9 \pm 8.1$ | $1056.3 \pm 80.1$ | $-818.2 \pm 832.9$ | $3218.9 \pm 281.4$ |
| **Ours** | $3554.6 \pm 275.6$ | $1891.8 \pm 197.0$ | $105.0 \pm 2.7$ | $\mathbf{1501.6 \pm 92.4}$ | $\mathbf{2868.1 \pm 95.3}$ | $\mathbf{3690.2 \pm 374.8}$ |
| **Ours-VAE** | $\mathbf{4078.8 \pm 295.1}$ | $2209.7 \pm 108.0$ | $88.4 \pm 9.4$ | $1334.7 \pm 148.9$ | $2400.5 \pm 423.1$ | $2159.4 \pm 1001.0$ |

Table 2: Learned policy performance for different imitation learning algorithms on **noisy** expert data, evaluated by using the mode of sampling from the policy distributions. The best results are marked as **red**.

The overall averaged scaled rewards for our method is about $0.907$ where the best baseline is $0.78$ for FAIRL. It is a more than $16.1\%$ relative improvement. Our method outperforms state-of-the-art baselines on all locomotion tasks except for Hopper where our method doesn't outperform PWIL. Moreover, the variational auto-encoder variant of our method gets the best scores on Walker2d and Hopper. In summary, our methods, including vanilla auto-encoder based and variational auto-encoder based variants, get better performance than other baselines comprehensively.

## 5.2 LEARNING FROM NOISY DEMONSTRATIONS

To show the robustness of our proposed AEIRL, we run our method and baselines on noisy expert demonstrations. The noise added to the expert data is a Gaussian noise $(0, 0.3)$ on every dimension of state action pairs on all tasks except for Ant and Humanoid. Since the dimension of state action pairs in Ant and Humanoid is much higher than other tasks, we choose $(0, 0.03)$ and $(0, 0.01)$ Gaussian noises added to the dataset respectively in order to lower the effects of the noise component. Figure 7 (See in Appendix A.3) depicts training curves for the ground truth rewards for different

methods on noisy expert data while Table 2 shows the final learned policy performance on MuJoCo tasks with ground truth episode return.

These results show that our method outperforms other state-of-the-art algorithms on all tasks except for Hopper and Swimmer, on which PWIL and GAIL wins respectively. Table 1 and 2 show that the noise component affects the performance of mimicking the expert behaviors. But our method could help to denoise the expert demonstrations and provide more robust reward signals for inducing the expert policies. The overall scaled rewards for our AEIRL is 0.795 where the best baseline is 0.542 for PWIL. Additionally, PWIL also has some capabilities to denoise the expert on part of the tasks. And other adversarial imitation learning methods, includes GAIL, WGAIL and FAIRL, are very sensitive to the noisy expert.

## 5.3 Analysis on Robustness of Our Method

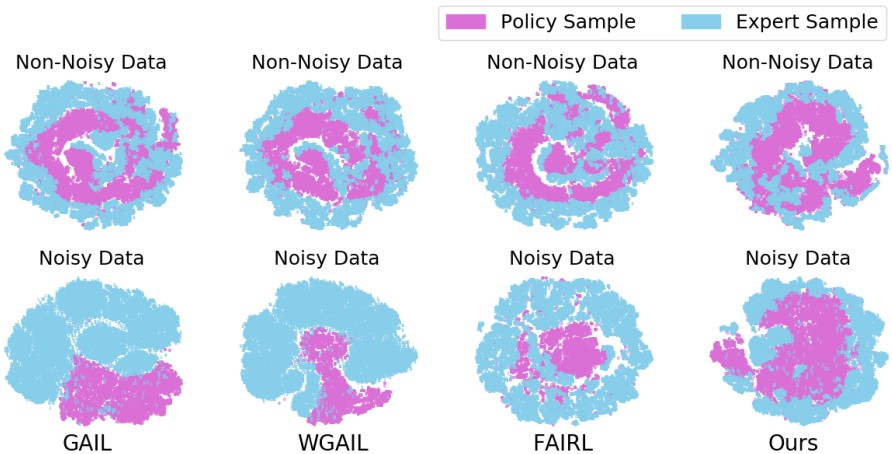

Figure 4: t-SNE visualization of latent representations of the reward function on Walker2d for different methods with hyper-parameter $preplexity = 30.0$. Top row: on non-noisy expert data, bottom row: on noisy expert data (Gaussian noise $(0, 0.5)$).

To compare the robustness of different reward signals of different methods, we use t-SNE visualization technique to visualize the latent space of the reward function network. For the discriminator based methods, we visualize the output of first hidden (middle) layer for instance. So the latent representations for both discriminator based and our auto-encoder based methods will show its robustness, especially to the corrupted expert data.

Since PWIL (Dadashi et al., 2021) considers an offline reward function without neural networks, we only compare our methods with other three IRL baselines. Figure 4 depicts that two sets of samples are more indistinguishable than other baselines both on clean and noisy expert data. It shows better robustness of our auto-encoder based reward function. More details for using t-SNE and results with varying $preplexity$ are shown in Appendix A.4

## 5.4 Ablation Studies

**Different Objectives** To analyze the major contributing factors of our method, we conduct the ablation studies based on different distribution divergences and different formulation of reward functions. Comparable performances would indicate that the major contributing factor is that we utilize the reconstruction error of an auto-encoder rather than the formulation of reward function and its corresponding objective. We formulate some other forms of surrogate reward functions based on Jensen-Shannon divergence (Goodfellow et al., 2014) to justify this hypothesis. Figure 9 (See in Appendix A.5) shows that three different formulations of surrogate reward signal achieve comparable results on almost all tasks which are all based on the vanilla auto-encoder. This shows the major contributing factor in our method is the auto-encoder based reward function which provides more informative learning signals, compared to the discriminator based reward function.

**Alternative Auto-Encoders** Our method is not limited to vanilla auto-encoders and we also take the variational auto-encoder as an alternative. We run variational auto-encoder based variant on MuJoCo tasks under the same setting with vanilla auto-encoder. Figure 10 (See in Appendix A.5) shows the training curves for variational auto-encoder based and vanilla auto-encoder based variants. Overall, the scaled rewards of variational auto-encoder based variant is $0.845$ and $0.727$ on clean and noisy expert respectively, which is comparable to the vanilla auto-encoder based method.

**Auto-Encoder based Reward Signal is Denser** A denser reward signal is beneficial for policy searching (Ng et al., 1999; Hu et al., 2020). Section 4.1 tells that our auto-encoder based reward function focuses on the full scale differences between the expert and generated samples. Thus, it provides more information to the agent compared to the discriminator based reward function. To justify this hypothesis, we want to illustrate the generated rewards for different trajectories in the whole trajectory space. Meanwhile more distinct rewards for different trajectories means the reward signal is denser and more informative. So we use the final learned reward function of our method and the baselines to score different trajectories for analysis.

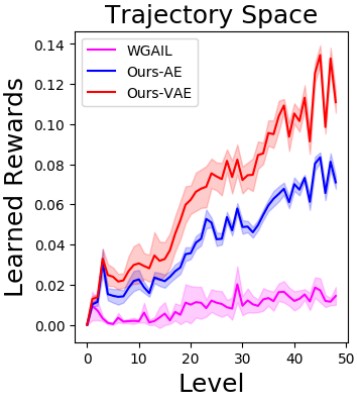

Figure 5: Scaled normalized rewards for different levels in the trajectory space on Walker2d.

We collect 50 levels of trajectories with different performances from random to the expert performance. This can be realized by saving trajectories every 100 iterations via a PPO. Since the trajectories with different performances have different lengths, we normalize the generated rewards with the length of trajectories. We also scale the generated rewards considering the expert samples are 1 and random policy samples are 0. So the scaled normalized rewards for different levels of trajectories reflect the denseness of a reward signal. And higher degree of distinction indicates it is a denser reward signal.

Figure 5 shows that the final learned scaled and normalized rewards for different levels of the trajectories. The higher level in the abscissa means higher performance of its trajectories. Note that learned rewards under our formulation are readily distinct at different levels when compared to learned rewards under WGAIL. Our much steeper curves indicates that the auto-encoder derived reward function provides a denser and more informative learning signal when compared to the discriminator based reward functions, which are all based on Wasserstein distance (Arjovsky et al., 2017).

**Stability of Our Method** The variance of gradients is computed via different mini-batches of training examples. Lower variance of gradients means smoother updating of the network (Faghri et al., 2020). Now that our auto-encoder based reward function is more informative and denser than the discriminator based ones, how about the training stability of our method? The relative variance of policy gradients to the weights of policy network shows the training stability for different methods. Figure 11 (See in Appendix A.6) shows that our method gets lower relative variance of gradients which means that the policy network training process can be much more stable.

## 6 CONCLUSIONS

This paper presents a succinct and robust adversarial imitation learning method based on Auto-Encoding (AEIRL). We utilize the reconstruction error of an auto-encoder as the surrogate reward function for reinforcement learning. The advantages of our method can be summarized into two aspects: (1) the auto-encoder based reward function focuses on the full scale differences between the expert and generated samples, which won't easily overfit to the expert data; (2) the auto-encoder denoises the expert data through the reconstruction process. Thus, the learned reward signal in our method is more robust and informative compared with the existing methods. Experimental results show that our methods achieve strong performances on locomotion tasks on both clean and noisy expert demonstrations. In the future, we want to further investigate our method in more realistic scenarios, such as autonomous driving.

## 7 REPRODUCIBILITY STATEMENT

Data and source code are attached in supplementary materials.

Experimental implementation details, including our methods and baselines, see in Appendix A.2.

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

# A   APPENDIX

## A.1   PSEUDO CODE

---

**Algorithm 1** Auto-Encoding Inverse Reinforcement Learning (AEIRL)

---

**Require:** Initial parameters of policy, auto-encoder $\theta_0$, $w_0$; Expert trajectories $D_E$.
**Ensure:**
1: **for** $i = 0$ to $N$ **do**
2:     Sample state-action pairs $(s_i, a_i) \sim \pi_{\theta_i}$ and $(s_E, a_E) \sim D_E$ with same batch size.
3:     Update $w_i$ to $w_{i+1}$ by decreasing with the gradient:

$$\mathbb{E}_{(s_i, a_i)}[\nabla_{w_i} 1/(1 + \text{AE}_{w_i}(s_i, a_i))] - \mathbb{E}_{(s_E, a_E)}[\nabla_{w_i} 1/(1 + \text{AE}_{w_i}(s, a))]$$

4:     Take a policy step from $\theta_i$ to $\theta_{i+1}$, using the TRPO update rule with the reward function $1/(1 + \text{AE}_{w_i}(s, a))$, and the objective function for TRPO is:

$$\mathbb{E}_{(s, a)}[-1/(1 + \text{AE}_{w_i}(s, a))].$$

5: **end for**

---

Algorithm 1 depicts the pseudo code for training AEIRL. The first step is to sample the state action pairs from expert demonstrations $D_E$ and the trajectories sampled by current policy with the same batch size. Then we update the auto-encoder by decreasing with the gradient with loss function Eq. 8. Finally, we update the policy assuming that the reward function is Eq. 6 which leads to an adversarial training paradigm. We runs these steps with $N$ iterations until the policy converges.

## A.2   IMPLEMENTATION

### A.2.1   TASKS

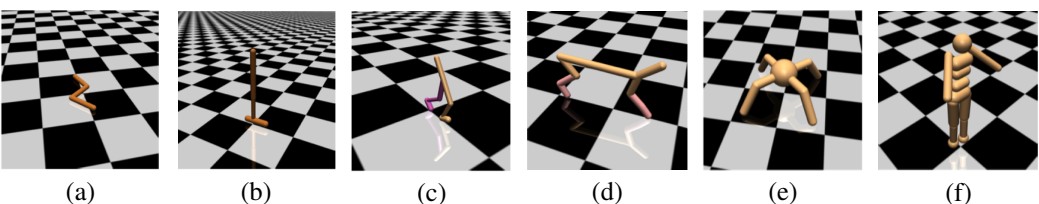

|  (a)  |  (b)  |  (c)  |  (d)  |  (e)  |  (f)  |

Figure 6: Illustrations for locomotion tasks we used in our experiments: (a) Swimmer; (b) Hopper; (c) Walker; (d) HalfCheetah; (e) Ant; (f) Humanoid.

The goal for all these tasks is to move forward as quickly as possible. These tasks are more challenging than the basic tasks due to high degrees of freedom. In addition, a great amount of exploration is needed to learn to move forward without getting stuck at local optima. Since we penalize for excessive controls as well as falling over, during the initial stage of learning, when the robot is not yet able to move forward for a sufficient distance without falling, apparent local optima exist including staying at the origin or diving forward slowly. Figure 6 depicts locomotion tasks' environments.

### A.2.2   BASIC SETTINGS

The expert trajectories are sampled with PPO on Walker2d, Hopper, and Swimmer, HalfCheetah and Ant with 25 trajectories. And we sample 320 trajectories on Humanoid with SAC to get higher expert performance since PPO is not good enough on Humanoid.

For each task, we normalize the state features and use the normalized features as the input for the auto-encoder. But we use the raw data input to compute the mean square error.

On Walker2d, Hopper, and Swimmer, we train these algorithms directly, while on HalfCheetah, Ant and Humanoid we use BC to pre-train the policy with 10k iterations. We test all the algorithms on

these six locomotion tasks with environment seed $0, 1, 2, 3, 4$, and using 10 rollouts to estimates the ground truth reward value to plot the training curves.

The maximum length for sampled trajectories is 1024. The discounted factor is $0.995$. Each iteration we update 3 times of policies and 1 time of reward function (discriminator or auto-encoder). While updating TRPO, we update the critic 5 times with learning rate 2e-4 every iteration.

### A.2.3  BASELINES

We use open source code, OpenAI baselines (Dhariwal et al., 2017), to get the performance of BC and the training iterations is 100k for all tasks.

GAIL (Ho & Ermon, 2016) borrows the GAN (Goodfellow et al., 2014) architecture to realize the inverse reinforcement learning process and it is applicable in high dimensional dynamic environments. We implement GAIL using open source code, OpenAI baselines (Dhariwal et al., 2017), with best tunned hyperparameters. We choose TRPO as the reinforcement learning algorithm.

WGAIL (Arjovsky et al., 2017; Ho & Ermon, 2016) utilizes the Wasserstein distance to train the policy and the reward function which can be more stable. WGAIL is implemented based on GAIL with corresponding objective function and reward signals as in (Arjovsky et al., 2017). We also choose TRPO as the reinforcement learning algorithm.

FAIRL (Ghasemipour et al., 2020) utilizes the forward KL divergence as the objective which shows competitive performances on MuJoCo tasks. FAIRL is implemented based on GAIL with corresponding objective function and reward signals as in (Ghasemipour et al., 2020). We also choose TRPO as the reinforcement learning algorithm.

PWIL (Dadashi et al., 2021) is an adversarial imitation learning method based on an offline similarity reward function. It is quite robust and easy to fine-tune the hyper-parameters of its reward function (only two hyper-parameters). PWIL is implemented based on open source code (Dadashi et al., 2021) with TRPO as the reinforcement learning step for fair comparison (original PWIL code uses D4PG as the RL algorithm).

### A.2.4  ARCHITECTURE OF OUR METHODS

**Policy Net:** the same with baselines, two hidden layers with 64 units, with tanh nonlinearities in between. The learning rate to update the policy is 0.01.

**Auto-Encoder Net:** both vanilla auto-encoders and variational auto-encoder, the same number of layers with the architecture of discriminator in GAIL and WGAIL, 4 layers. The two hideen layers are with 100 units, with tanh nonlinearities in between, the final output layer is an identity layer. And we normalize the state feature after the input layer and we use the raw state action input features to compute the mean square error. Learning rate for the auto-encoder is 3e-4.

### A.3  OMITTED EXPERIMENT RESULTS ON NOISY EXPERT DEMONSTRATIONS

To show the robustness of our AEIRL method, we conduct the experiments on noisy expert demonstrations. The noise component is a Gaussian distribution noise which is $(0, 0.3)$ for Walker2d, Hopper, Swimmer, HalfCheetah. Since the effects of the noise component performs more heavily on Ant and Humanoid, we choose $(0, 0.03)$ and $(0, 0.01)$ Gaussian distribution noise for Ant and Humanoid respectively. So under this setting, higher final ground truth rewards indicate better robustness of its corresponding algorithm.

Figure 7 shows the training curves for different imitation learning algorithms on the noisy expert demonstrations. On Walker2d, HalfCheetah, Ant, and Humanoid, our AEIRL clearly outperforms other baseline algorithms. On the other hand, on Swimmer GAIL achieves the best performance while on Hopper, PWIL achieves the best performance.

We scale the ground truth rewards of final policy performance for different methods which considers the expert trajectory's rewards to be 1 and random policy trajectory's rewards to be 0. And the averaged scaled ground truth rewards is $0.795$ while the best baseline is $0.542$ for PWIL. So we achieve $46.5\%$ relative improvement to PWIL. In summary, our method is more robust than other baselines.

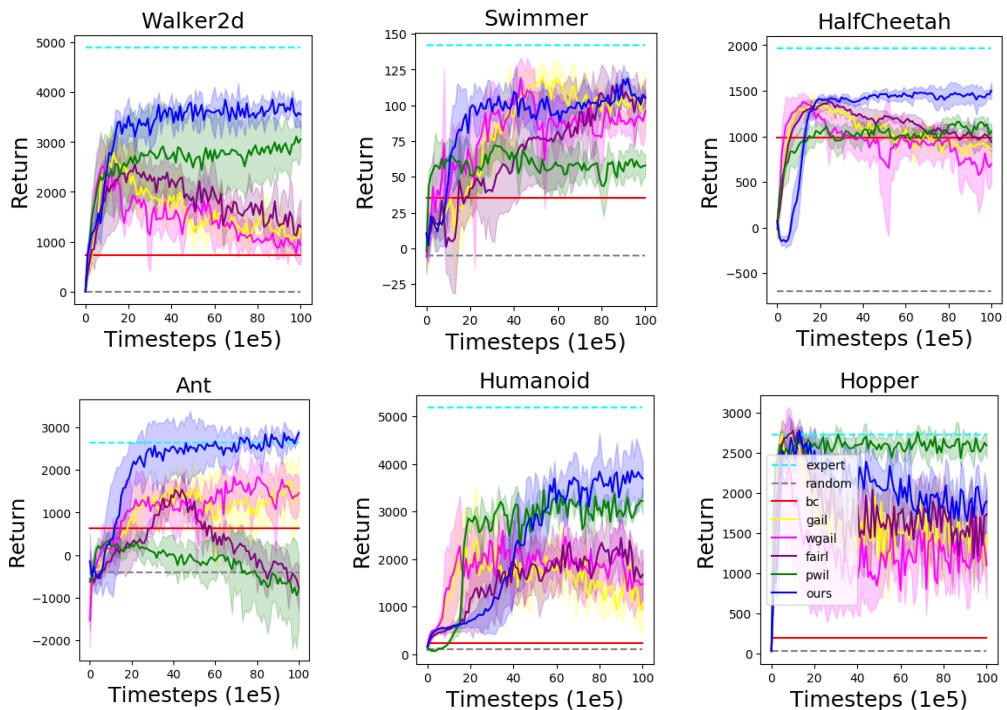

Figure 7: Mean and standard deviation return of the evaluation policy over 10 rollouts and 5 seeds learning from **noisy** expert demonstrations, reported every 100k timesteps. The return is in term of the environment's ground truth reward.

## A.4 DETAILS AND MORE RESULTS ON T-SNE VISUALIZATION

The auto-encoder based reward function learns the mapping from state action space to a scalar value, i.e., the reward. When the system of our method converges, the generated state action samples would be indistinguishable with the expert state action samples. For our auto-encoder based reward function, the latent representations between final policy samples and expert samples would also be indistinguishable.

To compare the robustness of different reward signals of different methods, we use t-SNE visualization technique to visualize the latent space of the reward function network. For the discriminator based methods, we visualize the output of first hidden (middle) layer for instance. And for our auto-encoder based reward function, we also visualize the output of first hidden (middle) layer in the net. So the latent representations for both discriminator based and our auto-encoder based methods will show its robustness, especially to the corrupted expert data.

We plot the t-SNE embedding results with SciKit-Learn tools (i.e., sklearn.manifold.TSNE function) with varying preplexity. The hyper-parameters for t-SNE are (other hyper-parameters are default with scikit-learn):

n_component = 2
early_exaggerationfloat = 12.0
learning_rate = 200.0
n_iterint = 1000
n_iter_without_progressint = 300
min_grad_normfloat = 1e-7
metric = 'euclidean'
init = 'pca'

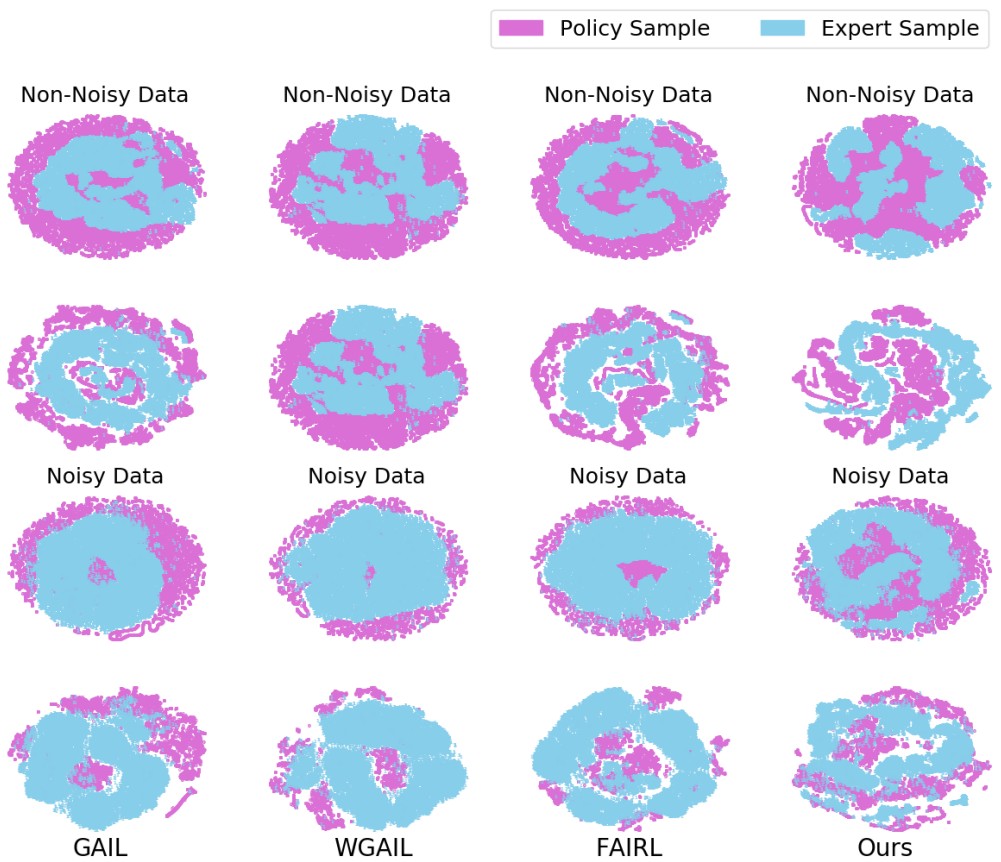

Figure 8: t-SNE visualization of latent representations of the reward function on Walker2d for different methods. Top row: on non-noisy expert data ($preplexity = 5.0$); Second row: bottom row: on non-noisy expert data ($preplexity = 50.0$); Third row: on noisy expert data ($preplexity = 5.0$); Bottom row: on noisy expert data ($preplexity = 50.0$).

Since PWIL (Dadashi et al., 2021) considers an offline reward function without neural networks, we only compare our methods with other three IRL baselines. Figure 8 depicts that two sets of samples are more indistinguishable than other baselines both on clean and noisy expert data. The top row of Figure 8 shows the embeddings with $preplexity = 5.0$ in the non-noisy setting. The second row shows the embeddings with $preplexity = 50.0$ in the non-noisy setting. The third row of Figure 8 shows the embeddings with $preplexity = 5.0$ in the noisy setting. The bottom row shows the embeddings with $preplexity = 50.0$ in the noisy setting. And the noise component is a Gaussian distribution $(0, 0.5)$. And in the main text, Figure 4 shows the default $preplexity = 30.0$ results.

With varying hyper-parameter preplexity, we can see the overlapped areas for policy and expert samples are greater for our method. In detail, we can observe that the policy samples for GAIL and WGAIL hardly appear in the outer ring with $preplexity = 30.0$. And FAIRL is a little better than GAIL and WGAIL. Our AEIRL gets maximum overlap between two sets of samples compared to these baselines. And in the noisy settings, the two sets of embeddings of GAIL and WGAIL are actually isolated with different "preplexity" while they are gobally clustered (greatly overlapped) only with $preplexity = 30.0$ for FAIRL. And the embeddings are greatly overlapped with varing preplexity for our method. So it reflects that our auto-encoder based reward function is more robust to both the clean and noisy data.

## A.5 ABLATION STUDIES

To illustrate the major contributing factors of our methods, we conduct the ablation studies with different forms of distribution divergences and other alternative auto-encoders.

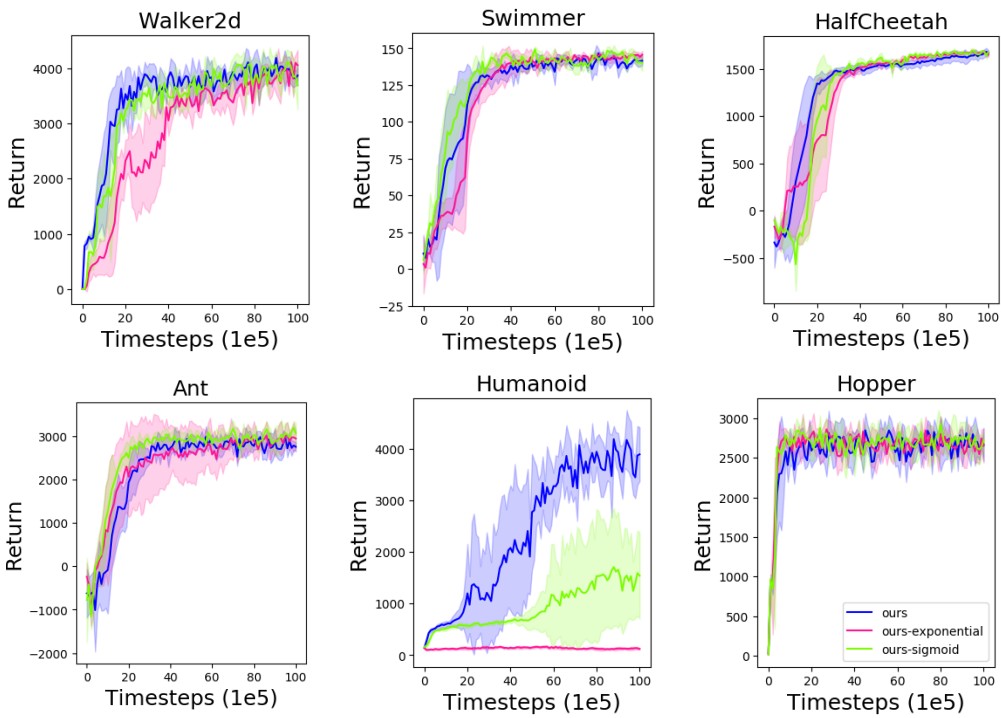

Figure 9: Mean and standard deviation return of the evaluation policy over 10 rollouts and 5 seeds learning from non-noisy expert demonstrations, reported every 100k timesteps. The return is in term of the environment's ground truth reward.

| Settings | Walker | Hopper | Swimmer | HalfCheetah | Ant | Humanoid |
|---|---|---|---|---|---|---|
| | $3897.8 \pm 177.8$ | $2706 \pm 219.7$ | $141.5 \pm 3.7$ | $\mathbf{1674.1 \pm 33.6}$ | $2715.7 \pm 93.1$ | $\mathbf{3892.3 \pm 522.9}$ |
| **Non-Noisy** | $\mathbf{4055.9 \pm 304.0}$ | $2580.3 \pm 186.4$ | $\mathbf{146.1 \pm 2.6}$ | $1660.6 \pm 26.3$ | $2930.4 \pm 236.8$ | $122 \pm 17$ |
| | $3589.1 \pm 427.4$ | $\mathbf{2770.5 \pm 66.8}$ | $143.1 \pm 3$ | $1670.6 \pm 20.4$ | $\mathbf{3062.9 \pm 282}$ | $2167 \pm 1212.1$ |
| | $3554.6 \pm 275.6$ | $\mathbf{1891.8 \pm 197.0}$ | $105.0 \pm 2.7$ | $\mathbf{1501.6 \pm 92.4}$ | $\mathbf{2868.1 \pm 95.4}$ | $\mathbf{3690.2 \pm 374.8}$ |
| **Noisy** | $3364.5 \pm 377.3$ | $1638.3 \pm 308.5$ | $\mathbf{107.8 \pm 9.6}$ | $1444.3 \pm 16.8$ | $2440.3 \pm 184.4$ | $123.2 \pm 23.1$ |
| | $\mathbf{3561.8 \pm 345.7}$ | $1671.7 \pm 126.7$ | $105.7 \pm 4.4$ | $1429.2 \pm 40.1$ | $2674.9 \pm 134.3$ | $2521.1 \pm 860.9$ |

Table 3: Learned policy performance for different imitation learning algorithms on **non-noisy** and **noisy** expert data, evaluated by using the mode of sampling from the policy distributions. The best results are marked as **red** respectively. Top: Ours; Middle: Ours-exponential; Bottom: Ours-sigmoid.

We conduct the experiments with another two forms of reward functions following the Jensen–Shannon divergence (Goodfellow et al., 2014) for comparison. Typically, JS divergence is defined as:

$$\mathrm{JS}(\mathrm{P}_r, \mathrm{P}_g) = \mathbb{E}_{x \sim \mathrm{P}_r}[\log(\frac{\mathrm{P}_r(x)}{\frac{1}{2}(\mathrm{P}_r(x) + \mathrm{P}_g(x))})] + \mathbb{E}_{x \sim \mathrm{P}_g}[\log(\frac{\mathrm{P}_g(x)}{\frac{1}{2}(\mathrm{P}_r(x) + \mathrm{P}_g(x))})] \qquad (10)$$

where $\mathrm{P}_r$ and $\mathrm{P}_g$ represents the real data distribution and the generated data distribution. The adversarial training procedure of auto-encoding inverse reinforcement learning will match the state action distribution of expert and generated samples.

In the first form of alternative reward functions, we consider the exponential of negative reconstruction error as: $\mathrm{P}_r(x)/\mathrm{P}_r(x) + \mathrm{P}_g(x)$. The corresponding Jensen-Shannon divergence for updating

the auto-encoder based reward function is:

$$\mathcal{L} = -\mathbb{E}_{(s,a)\sim D_E}[\log(\exp(-\mathrm{AE}_w(s,a)))] - \mathbb{E}_{(s,a)\sim \pi}[\log(1-\exp(-\mathrm{AE}_w(s,a)))] \quad (11)$$

$$=\mathbb{E}_{(s,a)\sim D_E}[\mathrm{AE}_w(s,a)] - \mathbb{E}_{(s,a)\sim \pi}[\log(1-\exp(-\mathrm{AE}_w(s,a)))] \quad (12)$$

Here, $w$ represents the parameters of the auto-encoder network. So its formulation of surrogate reward function is:

$$r_w(s,a) = -\log(1-\exp(-\mathrm{AE}_w(s,a))). \quad (13)$$

In this formulation, minimizing the Jensen-Shannon divergence between state action distribution of the expert and generated trajectories assigns low mean square errors to the expert samples and high mean square errors to other regions.

In the second formulation of reward functions, we consider the sigmoid output of the negative reconstruction error as: $\mathrm{P}_r(x)/\mathrm{P}_r(x)+\mathrm{P}_g(x)$. So it corresponding objective function for updating the auto-encoder is:

$$\mathcal{L} = -\mathbb{E}_{(s,a)\sim D_E}[\log(\mathrm{sigmoid}(-\mathrm{AE}_w(s,a))] + \mathbb{E}_{(s,a)\sim \pi}[\log(1-\mathrm{sigmoid}(-\mathrm{AE}_w(s,a)))] \quad (14)$$

where $w$ also represents the parameters of the auto-encoder network. And its reward function formulation is:

$$r_w(s,a) = \log(1-\mathrm{sigmoid}(-\mathrm{AE}_w(s,a))) \quad (15)$$

In experimental results, we note the first formulation as "ours-exponential" and the second formulation as "ours-sigmoid". Specifically, we don't add the entropy term to the reinforcement objective which already achieve competitive results on MuJoCo tasks.

These three forms of reward signals achieve comparable performance on all tasks except for Humanoid shown in Figure 9. This indicates that an auto-encoder based reward function is the major contributing factor to improve the performance on mimicking the expert rather than the different distribution divergences or forms of reward signal. However, the performance on Humanoid is quite different between these different forms of reward signals. It shows that the Wasserstein distance can be more robust to this task.

Our method is not limited to vanilla auto-encoders but also other alternative auto-encoders. To show the differences between our method and other alternative auto-encoders, we conduct the experiments with variational auto-encoder based reward function. The variational auto-encoder (Kingma & Welling, 2014) is a directed model that uses learned approximate inference. The key insight behind training variational auto-encoder is to maximize the variational lower bound $\mathcal{L}(q)$ of the log-likelihood for the training examples (Goodfellow et al., 2016):

$$\mathcal{L}(q) = \mathbb{E}_{z\sim q(z|x)}\log p_{\mathrm{model}}(z,x) + \mathcal{H}(q(z|x)) \quad (16)$$

$$= \mathbb{E}_{z\sim q(z|x)}\log p_{\mathrm{model}}(x|z) - D_{\mathrm{KL}}(q(z|x)\|p_{\mathrm{model}}(z)) \quad (17)$$

$$\leq \log p_{\mathrm{model}}(x) \quad (18)$$

where $z$ is the latent representation for data points $x$. And $p_{\mathrm{model}}$ is chosen to be a fixed Gaussian distribution $\mathcal{N}(0,I)$. In Eq. 16, we recognize that the second entropy term encourages the variational posterior to place high probability mass on $z$ values as many as possible, rather than collapse to a single most likely value point. When we use an variational auto-encoder based reward function in our method, the latent representations of the expert samples can be widely distributed on a Gaussian distribution. Therefore, matching the expert and generated state action distributions in this latent space leads to more diverse behaviors inducing. Thus, the reward signal would be more informative.

Training the variational auto-encoder is still a adversarial process. We notice that the first term in Eq. 17 is the reconstruction log-likelihood found in traditional auto-encoders. So the objective for training the variational auto-encoder is:

$$\mathcal{L} = \mathbb{E}_{(s,a)\sim D_E}[r_w(s,a) + D_{\mathrm{KL}}(p(z|(s,a)),p_{\mathrm{model}}(z))]$$
$$- \mathbb{E}_{(s,a)\sim \pi_\theta}[r_w(s,a) + D_{\mathrm{KL}}(p(z|(s,a)),p_{\mathrm{model}}(z))] \quad (19)$$

where the choice for $r_w(s,a)$ is the same with Eq. 6 in Section 4.2. Therefore, low mean square errors of the variational auto-encoder ensure high reward values for state action pairs and vice versa.

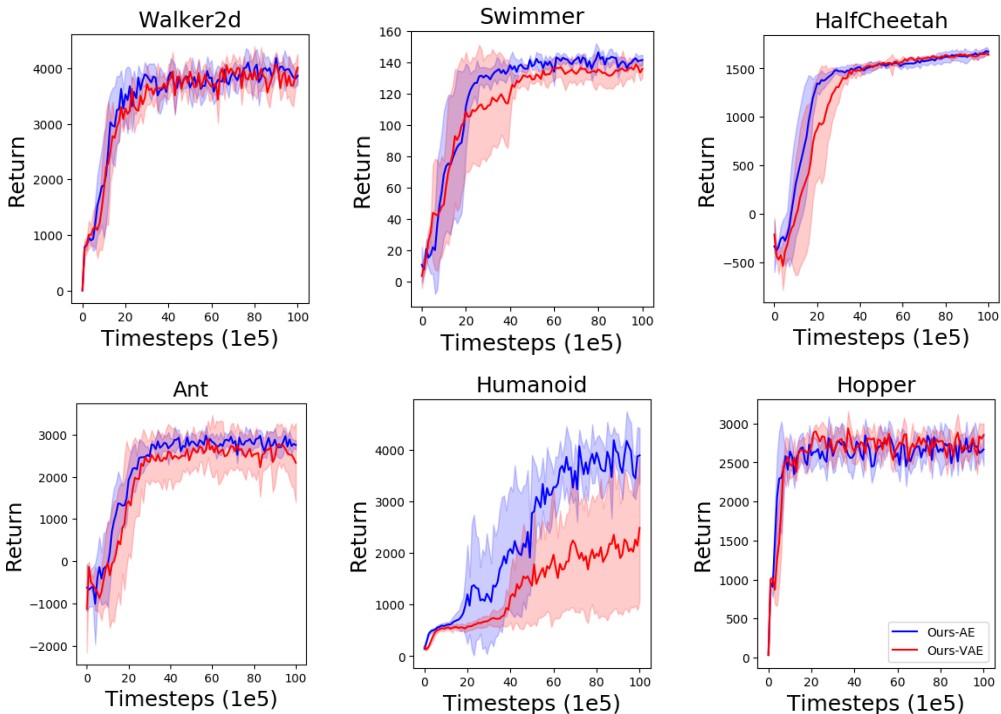

Figure 10: Mean and standard deviation return of the evaluation policy over 10 rollouts and 5 seeds learning from non-noisy expert demonstrations, reported every 100k timesteps. The return is in term of the environment's ground truth reward.

When minimizing this objective, the variational auto-encoder based reward function can still assign high values near the expert demonstrations and low values to other regions which is the same as in typical auto-encoders. On the other hand, optimizing this objective is restricting the latent distribution for the expert to a fixed Gaussian distribution $p_{model}(z))$ and maximizing the KL distance between the generated samples and the same fixed Gaussian distribution. With the bi-level optimization process 4, we can induce the expert policy with more diverse behaviors from the expert demonstrations which addresses the problem of mode collapse in GAIL. And a more informative reward signal would be yield.

Figure 10 shows the training curves for vanilla auto-encoder based variant and varational auto-encoder based variant on non-noisy expert demonstrations. The performances for these two variants are comparable on almost all tasks while vanilla auto-encoder based method is better than varational auto-encoder based variant on Humanoid. It indicates that the variational auto-encoder can be an alternative reward function formulation.

## A.6 ANALYSIS ON STABILITY OF OUR METHODS

The variance of gradients is computed via different mini-batches of training examples. Lower variance of gradients means smoother updating of the network (Faghri et al., 2020). Now that our auto-encoder based reward function is more informative and denser than the discriminator based ones, how about the training stability of our method? To show the training stability of different methods, we record the norm of weights of the policy network and the variance of policy gradients. The relative variance of policy gradients shows the training stability for different methods.

While we utilize Wasserstein distance in our method, we compare our vanilla auto-encoder based variant and variational auto-encoder based variant with WGAIL. It could depict the training stability with our newly proposed auto-encoder based reward formulation. Figure 11 shows the training stability of our method compared to the baseline WGAIL. It depicts that our method can be more stable and the policy network updating is smoother.

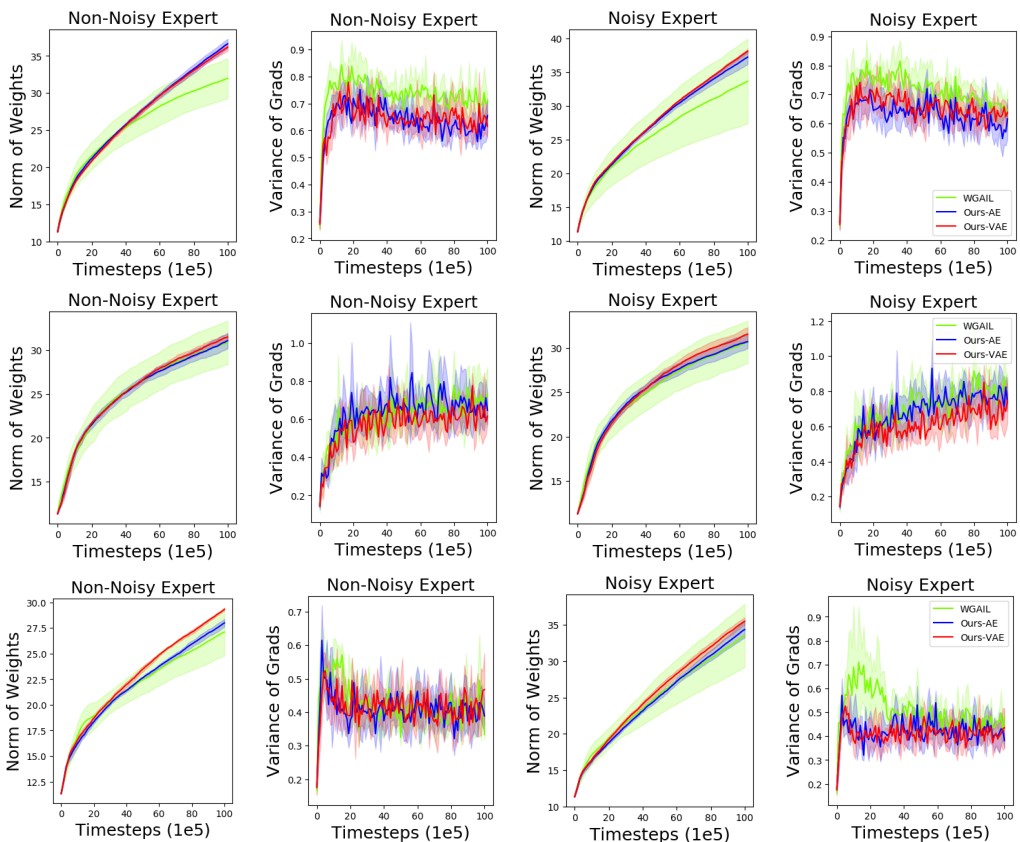

Figure 11: More stable training procedure: norm of weights for the policy network and the corresponding variance of policy gradients for non-noisy and noisy expert data. Top: Walker2d; Middle: Swimmer; Bottom: Hopper.

## A.7 ADDITIONAL EXPERIMENTS ON BETTER SYNTHESIS EXPERTS

| Task | Walker | Hopper | HalfCheetah | Ant |
|------|--------|--------|-------------|-----|
| Expert | 6121.7 | 3548.0 | 8879.1 | 4177.0 |
| GAIL | $2933.4 \pm 392.6$ | $2807.1 \pm 196.1$ | $3107.4 \pm 1138.9$ | $1866.9 \pm 688.8$ |
| WGAIL | $2692.5 \pm 746.7$ | $2850.3 \pm 198.8$ | $3502.2 \pm 1411.0$ | $1964.6 \pm 575.1$ |
| FAIRL | $2425.0 \pm 216.2$ | $2630.4 \pm 263.8$ | $1655.9 \pm 591.6$ | $2107.2 \pm 324.7$ |
| PWIL | $1414.8 \pm 1033.8$ | $\textbf{\textcolor{red}{2969.5} \pm \textcolor{red}{86.7}}$ | $5608.8 \pm 411.0$ | $1661.2 \pm 397.8$ |
| **Ours** | $\textbf{\textcolor{red}{4036.8} \pm \textcolor{red}{159.8}}$ | $2939.2 \pm 183.4$ | $\textbf{\textcolor{red}{5775.1} \pm \textcolor{red}{644.8}}$ | $\textbf{\textcolor{red}{3104.1} \pm \textcolor{red}{95.4}}$ |

Table 4: Learned policy performance for different imitation learning algorithms on **non-noisy** expert data, evaluated by using the mode of sampling from the policy distributions. The best results are marked as **red**.

To get more convincing results, we run the experiments on better synthesis experts. We collect the expert demonstration with 16 trajectories each task via D4PG (Barth-Maron et al., 2018).

Figure 12 depicts the training curves of ground truth rewards for different algorithms. The training curve shows the imitation performance learning from non-noisy expert data. Table 4 shows the final learned policy performance on MuJoCo tasks with ground truth episode return.

The overall results are similar with what we have done in the main text. On Walker2d, HalfCheetah and Ant, our method gets the best performance. PWIL gets the best performance and our AEIRL is on par with PWIL on Hopper.

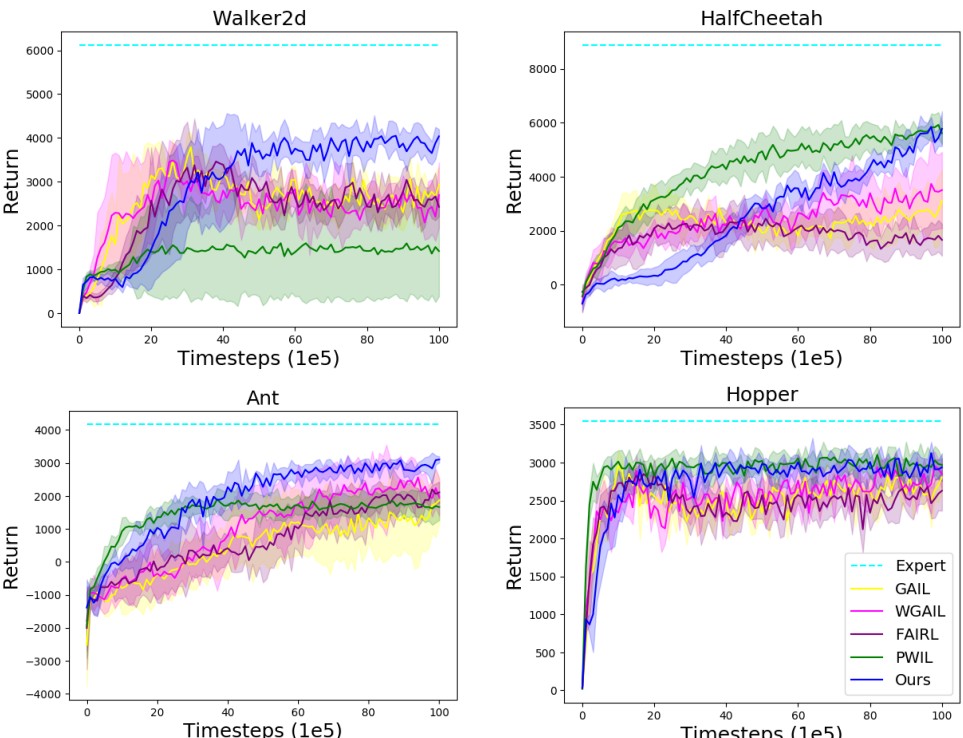

Figure 12: Mean and standard deviation return of the evaluation policy over 10 rollouts and 5 seeds, learning from **non-noisy** expert demonstrations, reported every 100k timesteps. The return is in term of the environment's ground truth reward.

We take the random and expert trajectory as 0 and 1 respectively to compute the scaled final rewards for different methods. The overall averaged scaled rewards for our method is about 0.729 where the best baseline is 0.546 for PWIL. It is a more than 33.7% relative improvement. So our method outperforms the other state-of-the-art baselines mostly.

