# OpenReview forum: "Auto-Encoding Inverse Reinforcement Learning"
_ICLR.cc/2022/Conference — ICLR 2022 Submitted_

### Official Review · Reviewer_4gyw · 2021-10-25

**Correctness:** 3
**Technical Novelty And Significance:** 3
**Empirical Novelty And Significance:** 2
**Recommendation:** 5
**Confidence:** 3

**Main Review:**

The paper is an interesting approach to IRL, using minimax games and autoencoding to enhance training.  While the approach appears novel (IRL is not my main area), the experimental results, while generally positive, are a little mixed in terms of which approach is preferred (especially VAE vs non-VAE).

The empirical results on noise-free data are impressive (as are some of the noisy results), but I would have liked to have seen more explanation on why their VAE-based approach outperforms non-VAE on some tasks but not others. I.e., which one should be chosen based on the task?

I don't understand the significance of Figure 4.  In what way are your approach's latent representations "more indistinguishable" from those of the others?  In particular, I don't see any real difference on the non-noisy data.  Can you quantify this somehow?

I read the authors' response, but it didn't compel me to change my score.

**Summary Of The Paper:**

The authors present an architecture that utilizes autoencoding as an approach to inverse reinforcement learning.  They present a mathematical formalism, pseudocode, and some empirical analyses on MuJoCo tasks.

**Summary Of The Review:**

The paper is an interesting approach to IRL, using minimax games and autoencoding to enhance training.  While the approach appears novel (IRL is not my main area), the experimental results, while generally positive, are a little mixed in terms of which approach is preferred (especially VAE vs non-VAE).

---

> ### Author Response · Authors · 2021-11-16
> **Response to Reviewer 4gyw and Changes**
>
> Thank you for your valuable feedback and suggestions!
>
>
>
> **Q1: The experiments are a little mixed in terms of which approach is preferred (especially VAE vs non-VAE).**
>
> A1: It is reasonable to use either VAE or non-VAE approach. The experiments with VAE based variant are only to proof the importance of the encoding-decoding process. And their performances are comparable.
>
>
>
>
> **Q2:  In what way are your approach's latent representations "more indistinguishable" from those of the others, especailly on the non-noisy data.**
>
> A2: We consider that "more indistinguishable" as larger overlapping areas between the expert and policy sample embeddings. In the non-noisy setting, the policy samples for GAIL and WGAIL hardly appear in the outer ring. And FAIRL is a little better than GAIL and WGAIL, and it is close to our method. Our AEIRL gets maximum overlap between two sets of samples compared to other baselines.
> And in the noisy setting, the two sets of embeddings of GAIL and WGAIL are actually isolated.  And the two sets of sample embeddings are gobally clustered for FAIRL but still not much overlapped. On the other hand, the embeddings are greatly overlapped for our AEIRL. So it indicates that our auto-encoder based reward function is more robust on both the clean and noisy data.
>
> Thank you,
>
> Authors

---

### Official Review · Reviewer_Ui5q · 2021-10-30

**Correctness:** 3
**Technical Novelty And Significance:** 2
**Empirical Novelty And Significance:** 2
**Recommendation:** 3
**Confidence:** 3

**Main Review:**

Pros:
1. Auto-encoder based reward function was proposed to denoise the expert data through the encoding-decoding process.
2. The experiments on the MoJoCo showed that the proposed method outperformed state-of-the-art imitation learning on both clean and noisy expert demonstrations.
3. Ablation studies showed that different reward functions and loss functions also received good performance.

Cons:
1. Theoretical justification to support the idea of auto-encoder based surrogate reward function was missing.
2. This method was to minimize the Wasserstein distance between the state action distribution of the policy and the expert demonstrations. But, there was not explanation on Lipschitz constraint or weight clipping.
3. This paper mentioned WGAIL (Arjovsky et al., 2017) many times, but Wasserstein GAN did not use reinforcement learning or imitation learning.
4. Need theoretical discussion about using the variational auto-encoder based reward function and the Jensen-Shannon divergence to replace auto-encoder and Wasserstein distance, respectively.
5. Lack of suitable experimental results to support that auto-encoder based reward function was more informative and denser than the discriminator based one.

**Summary Of The Paper:**

This paper presented the auto-encoding inverse reinforcement learning where the reward function is recovered from expert demonstrations in inverse RL problem. The reward was represented as a surrogate function which used the auto-encoder with the metric of reconstruction error.

**Summary Of The Review:**

A new auto-encoder based reward function was presented. Theoretical and experimental justifications were insufficient.

---

> ### Author Response · Authors · 2021-11-16
> **Response to Reviewer Ui5q and Changes**
>
> Thank you for your valuable feedback and suggestions!
>
> **Q1: Theoretical justification to support the idea of auto-encoder based surrogate reward function was missing.**
>
> A1: AEIRL shares a similar theortical framework with GAIL where the auto-encoder based reward function can be seen as a generalized discriminator. The reason is that our auto-encoder reward function shares the same set of properties with GAIL's discriminator, and GAIL is proven to converge under certain conditions. When the discriminator in GAIL satisifies: (1) the output of the discriminator based reward is bounded; (2) encourage the expert samples to have higher rewards and penalize other regions; (3) the objective function is properly regularized (to be strongly concave) with respect to the discriminator, [2] theoretically justifies that GAIL would converge globally. For our AEIRL, the surrogate reward funciton is also bounded and it also provides high values to the region near the expert demonstrations and penalizes other regions. Condition (3) is an assumption for theoretical convenience, since it doesn't hold for multi-layer neural network discriminator in GAIL. Here we make the same assumption. So similiarly, our AEIRL shares the same theoretical properties with GAIL which means that our AEIRL can globally converge under these conditions.
>
>
>
> **Q2: The method is to minimize the Wasserstein distance between the state action distribution of the policy and the expert demonstrations. But, there was not explanation on Lipschitz constraint or weight clipping.**
>
> A2: Wasserstein distance is a distance function between probability distributions on a given metric space. WGAN/WGAN-GP chooses the metric functions to be a class of all 1-Lipschitz continuous functions while we choose it as a class of neural networks for computational convenience. So weight clipping / gradient penalty (typically in WGAN / WGAN-GP) is optional for our method and we did not use it in our method. Emprically, [3] tells that the performances for adversarial imitation learning with regularization (or not) for the reward function are actually on par.
>
>
>
> **Q3: This paper mentioned WGAIL (Arjovsky et al., 2017) many times, but Wasserstein GAN did not use reinforcement learning or imitation learning.**
>
> A3: It is a mistake to cite WGAN (Arjovsky et al., 2017) only. Instead, we revised to cite WGAN and original GAIL paper both for WGAIL. WGAIL is a combination of WGAN and GAIL. It is widely used as an baseline in adversarial imitation learning field such as in [4]. We introduce the implementation details for WGAIL in Appendix A.2.3.
>
>
>
> **Q4: Need theoretical discussion about using the variational auto-encoder based reward function and the Jensen-Shannon divergence to replace auto-encoder and Wasserstein distance, respectively.**
>
> A4: Our use of VAE, and JS divergence in place of the original AE or W-distance is aim to demonstrate our AEIRL works under a wide range of auto-encoder variants, and a wide range of distance functions. We don't claim exact equivalences between VAE-based and AE-based variants or between the JS divergence and Wasserstein distance. We conduct the ablation studies with JS divergence and VAE-based reward function only to proof the importance of using the auto-encoder in our method.
>
>
>
> **Q5: Lack of suitable experimental results to support that auto-encoder based reward function was more informative and denser than the discriminator based one.**
>
> A5: We have shown the final recovered rewards for different levels of trajectires in Figure 5. And our much steeper curves indicate a denser and more informative reward signal compared to the discriminator based one. (To justify that our auto-encoder based reward function is denser and more informative, we illustrate the generaed rewards for different performances of trajectories in the whole trajectory space. And more distinct rewards for different level of trajectories means the reward signal is denser and more informative.)
>
> Thank you,
>
> Authors
>
> [2] Z. Guan et al., When will Generative Adversarial Imitation Learning Algorithms Attain Global Convergence. AISTATS'2021.
>
> [3] M. Orsini et al., What Matters for Adversarial Imitation Learning. NeurIPS'2021.
>
> [4] T. Xu et al., Errors Bounds for Imitation Policies and Environments. NeurIPS'2020.

---

### Official Review · Reviewer_vWWw · 2021-11-03

**Correctness:** 4
**Technical Novelty And Significance:** 3
**Empirical Novelty And Significance:** 4
**Recommendation:** 8
**Confidence:** 4

**Main Review:**

Paper Strengths:
* AEIRL intuitively provides several benefits. (1) it denoises potentially noisy expert demonstrations, whereas prior inverse RL methods struggle to work with noisy expert demonstrations. (2) the auto-encoder also provides a more informative reward signal whereas the binary classification objective in works like GAIL, can easily result in overfitting.
* The method seems easy to implement only requiring changing the discriminator to an auto-encoder.
* The main experiments of the paper are thorough. In both non-noisy and noisy demonstrations they compare on 6 standard benchmark tasks against 5 baselines and two versions of their method, all using 5 seeds.
* Their method performs convincingly better in the non-noisy setting and mostly better in the noisy setting.
* There are extensive analyses and ablation studies. I especially found the study of the reward signal in Figure 5 to be informative. This adds qualitative evidence to AEIRL learning a more dense reward function.

Paper Weaknesses:
* Why does AEIRL perform relatively better on non-noisy expert demonstrations compared to noisy expert demonstrations? The motivation of the method is that it can work well from noisy inputs, yet the method is outperformed in several of the noisy settings and has a large performance drop. This is the opposite of what I would expect.
* While the reward is intuitive, can it recover a true reward function, or is the reward coupled with the policy like with GAIL?
* This method is hard to scale to higher dimensional tasks like image-based control. AEIRL must learn a decoder to reconstruct observations. The reconstruction error could be less reliable as a reward signal in these higher dimension tasks since learning the decoder is harder.
* What happened on the Ant task with noise? Many of the methods went from very positive rewards to negative rewards.
* The numbers in Tables 1 and 2 are too close together, making some of them hard to read.

**Summary Of The Paper:**

This paper proposes the Auto-Encoding Inverse Reinforcement Learning (AEIRL) method for better imitation learning, especially in the presence of noisy expert demonstrations. The paper’s key insight is that auto-encoders can eliminate the effects of noise from expert demonstrations and provide a more stable reward signal based on the auto-encoder error. The experiments feature extensive analyses across standard benchmarks compared to a large number of baselines.

**Summary Of The Review:**

The method is intuitive and performs strongly across in an extensive imitation learning evaluation. Their analyses are also convincing that their. I, therefore, recommend acceptance.

---

> ### Author Response · Authors · 2021-11-16
> **Response to Reviewer vWWw and Changes**
>
> Thank you for your valuable feedback and suggestions!
>
> **Q1: Why does AEIRL perform relatively better on non-noisy expert data compared to noisy expert data?**
>
> A1: AEIRL outperforms other baselines on most tasks, and is comparable to the best baseline on Swimmer and Hopper. For example, on Swimmer, AEIRL gets only 1.4 rewards less than the best one. On the other hand, with the growth of the dimension for state action space, the superiority of our AEIRL would be greater due to the encoding-decoding process. For example, our method achieves the largest improvement on the highest dimensional task, Humanoid, compared to the GAIL baseline. We hypothesis that the dimension of Hopper and Swimmer are too small to show the efficacy of the denoising capability of our auto-encoder based method. So our AEIRL doesn't get the best performance on these two tasks in the noisy setting but on par with the best one.
>
>
>
> **Q2: While the reward is intuitive, can it recover a true reward function, or is the reward coupled with the policy like with GAIL?**
>
> A2: The reward function in our AEIRL is coupled with the policy which is similar to GAIL. And it can be simplely adapted to recovering a true reward function with other divergences (like in AIRL).
>
>
>
> **Q3: This method is hard to scale to higher dimensional tasks like image-based control since training the decoder of AEIRL is hard.**
>
> A3: Yes, we agree with you. In our experiments, we can observe that with the growth of the dimension of the state action space (from Swimmer to Humanoid), the relative improvements for our AEIRL compared to other baselines are greater. So we think it might indicates that there might be more struggles for other IRL methods to be applied into higher dimensional tasks like image-based control. However, we might also be able to use contrastive learning to help our AEIRL or other IRL methods to get good performances on these tasks in the furture.
>
>
>
> **Q4:  The performance with noisy expert on Ant seems to be strange?**
>
> A4: The effects of the noise component grows with higher dimension of the state-action space since we added a specific Gaussian noise distribution to every dimension of the samples in our experiments.  On Ant, the noise which we chose is too strong with (0, 0.3) Gaussian distribution. We revised the noise to (0, 0.03) Gauss which fix the many negative reward issue. And AEIRL still gets the best performance compared to other baselines. The paper has updated to reflect this (Section 5.2).
>
>
>
> **Q5: The numbers in Tables 1 and 2 are too close together, making some of them hard to read.**
>
> A5: We revised the format of the tables and it seems to be better now.
>
> Thank you,
>
> Authors

---

> > ### Comment · Reviewer_vWWw · 2021-11-18
> > **Post-Response**
> >
> > Thank you for the response.
> >
> > **Re A1,A3**: The authors state AEIRL scales better to high dimensional tasks than baselines. However, this contradicts the decoder and reconstruction objective of AEIRL intuitively being harder to scale to higher dimension tasks. Reconstructing higher dimension observations is harder which should provide a worse learning signal for the policy. Then how is it possible that AEIRL can empirically scale better than baselines to higher dimension tasks?
> >
> > **Re A2**: Given the method does not recover a true reward function, "inverse reinforcement learning" (IRL) does not belong in the title or method name since IRL refers to extracting a reward and not just a policy as the method currently does.
> >
> > Thank you for updating the table formatting and fixing the Ant experiment setting.

---

> > > ### Author Response · Authors · 2021-11-18
> > > **Response to Reviewer vWWw**
> > >
> > > Thanks for your response and suggestions!
> > >
> > > **Re Point 1:** We agree with that when the dimension for the state-aciton sapce grows, it would be harder for our method to train the decoder. Therefore, the reward signal would be worse on higher dimensional tasks. However, for the baselines (i.e., GAIL), the discriminator would focus much more on the minor differences between the expert and agent samples with the growth of the dimensionality, especailly with image based states. It is also harder for GAIL to be apllied into high dimensional spaces. When we say our method is better in high dimensional space, we meant we are "realatively" better than the baselines, because of the denser signal in the AE. This is also consistent with our empirical observations, where our method on Humanoid has the largest relative improvement while HalfCheetah has the lowest relative improvement.
> > >
> > > **Re Point 2:** Thank you for pointing out the issue that our method actually learns a coupled reward function with the policy. We recognize this problem and we think the title "Auto-Encoding Adversarial Imitation Learning" might be more appropriate for our paper. We will revise this issue.
> > >
> > > Thank you,
> > >
> > >
> > > Authors

---

### Official Review · Reviewer_jqo8 · 2021-11-03

**Correctness:** 4
**Technical Novelty And Significance:** 3
**Empirical Novelty And Significance:** 3
**Recommendation:** 6
**Confidence:** 4

**Main Review:**

Pros:
* The use of auto-encoder and reconstruction error for formulating reward functions is novel.
* The auto-encoder is flexible and has many choices in the literature, and I appreciate the authors' effort of trying different variants on that in ablation study.
* The ablation study on distance function for inverse RL loss function is also impressive, which shows the importance of using auto-encoder.
* I appreciate the effort of visualizing the latent representation of the reward model, though with some concerns on the plot. Please see details below.

Some questions and concerns:
* I wonder why using TRPO as the baseline agent, especially when obtaining the expert agent.
  * It's well-known that the performance of learning a reward model (and also a police) is affected by how good the expert agent is, so I wonder if you have tried some more recent and better agents in the literature.
  * For example, the PWIL paper shows a great agent with much higher score on Humanoid (nearly 9000) and results in the imitating agent to achieve 7000+ scores, and their agent was trained by D4PG.
* The description of agent and reward model (and/or auto-encoder) architecture is missing. Please add it for completeness and reproducibility.
* As it has been widely discussed in the literature, there are too many misuses of t-SNE due to the lack of misunderstanding hyper-parameters and possibly more details. I generally recommend the authors to provide more details when doing the t-SNE plots. For example, the authors can provide the value of `perplexity` that used for the figure (you can do that in appendix), and possibly plot more than one plots by varying `perplexity`.
  * Also, please provide details on which latent space is plotting, for both auto-encoder and discriminators.
* *Question*: is it correct to say, a well-trained auto-encoder in the loop (e.g, in Fig. 1) can encode expert samples better with lower reconstruction error, while it doesn't capture the features of generated samples and result in higher error?
* *Question*: is it possible to provide theoretical analysis on why the proposed method can minimize the distance between the state-action pair distributions of expert and agent samples, or other related similarity / distance metrics?

**Summary Of The Paper:**

This paper proposes to use auto-encoder for inverse reinforcement learning. The main goal of using the auto-encoder, is to use it as the reward function, which takes the auto-encoder reconstruction error to provide reward signals for the agent. The authors claim that such approach provides more informative signal than existing works, especially adversarial imitation learning approaches. Their experiments demonstrate that this method has a better performance over other baselines, and more robust to demonstration noise.

**Summary Of The Review:**

The authors provide a good insight on how the adversarial imitation learning algorithms fail. The sensitivity to minor differences between expert and agent samples does impede the learning of agent because it may be pushed to learn how to imitate the details instead of other more important features, and thus less robust to the noise of data. Given that, the authors introduce auto-encoder to handle this issue, and the experiments confirms the claim by showing its better performance over other baselines.

However, I still have some concerns regarding the writing, mostly the lack of important details. The experiments may need to conduct in another round with a more recent and powerful RL agent, which should be easily available now. I also hope to see if the authors can evaluate their approach in more complex and/or real-world environments. The theoretical understanding of the proposed approach is also worth further investigating.

---

> ### Author Response · Authors · 2021-11-16
> **Response to Reviewer jqo8 and Changes**
>
> Thank you for your valuable feedback and suggestions!
>
> **Q1: Why using TRPO as the baseline agent, especially when obtaining the expert agent?**
>
> A1: The expert demonstrations for Humanoid in our paper is sampled via SAC  since PPO/TRPO performs not good enough on this task, and other expert data are sampled with PPO (Details See in Appendix A.2.2 ). Thus, our method is not sensitive to how the expert data is collected. To get more convincing results, we conduct another round of experiments with a more recent and powerful D4PG agent in Appendix A.7, we find that on these datasets, our method achieves 0.729 scaled averaged rewards overall, and the best baseline is PWIL achieves 0.546 on clean datasets. We provide the new dataset and code for reproducing these results in supplementary materials.
>
>
> **Q2: There are missing some implemetation details, i.e., agent and reward model description?**
>
> A2: The description of agent and reward model see in Appendix A.2.4. We also added more details for implement our algorithm  in Appendix A.2 while code/data see in supplementary materials.
>
>
>
> **Q3: Misuses of t-SNE? And lack of hyper-parameters and possibly more details, such as which latent space is poltted?**
>
> A3: The latent space of auto-encoder/discriminator which we plot is the output of the first hidden layer (middle) of the net. And we added more details (for using t-SNE, all hyper-parameters including: learning_rate, n_iterint etc. ) and results on varying preplexity from 5.0, 30.0 to 50.0 (Also see in updated Appendix A.4). Our method is consistently better than the baselines under different perplexities. The original t-SNE plot in the main text is based on preplexity=30.0. Our AEIRL gets maximum overlap between two sets of samples compared to other baselines.  It indicates that our auto-encoder based reward function is more robust.
>
> **Q4: I hope to see if the authors can evaluate their approach in more complex and/or real-world environments.**
>
> A4: We agree with that imitation in more complex environment is important. We think it is worth investigating more about our AEIRL combining with other techniques such as contrastive learning etc. in the furture.
>
>
> **Question #1: is it correct to say, a well-trained auto-encoder in the loop (e.g, in Fig. 1) can encode expert samples better with lower reconstruction error, while it doesn't capture the features of generated samples and result in higher error?**
>
> Answer #1: Yes.
>
>
> **Question #2: is it possible to provide theoretical analysis on why the proposed method can minimize the distance between the state-action pair distributions of expert and agent samples, or other related similarity / distance metrics?**
>
> Answer #2: AEIRL shares a similar theortical framework with GAIL where the auto-encoder based reward function can be seen as a generalized discriminator. The reason is that our auto-encoder reward function shares the same set of properties with GAIL's discriminator, and GAIL is proven to converge under certain conditions. When the discriminator in GAIL satisifies: (1) the output of the discriminator based reward is bounded; (2) encourage the expert samples to have higher rewards and penalize other regions; (3) the objective function is properly regularized (to be strongly concave) with respect to the discriminator, [1] theoretically justifies that GAIL would converge globally. For our AEIRL, the surrogate reward funciton is also bounded and it also provides high values to the region near the expert demonstrations and penalizes other regions. Condition (3) is an assumption for theoretical convenience, since it doesn't hold for multi-layer neural network discriminator in GAIL. Here we make the same assumption. So similiarly, our AEIRL shares the same theoretical properties with GAIL which means that our AEIRL can globally converge under these conditions.
>
> Thank you,
>
> Authors
>
> [1] Z. Guan et al., When will Generative Adversarial Imitation Learning Algorithms Attain Global Convergence. AISTATS'2021.

---

### Decision · Program_Chairs · 2022-01-20

**Decision:**

Reject

**Comment:**

This work addresses the problem of learning representations from noisy expert demonstrations in in adversarial imitation learning. The authors build on top of GAIL, which utilizes a discriminator to model a "pseudo"-reward from demonstrations. In this work, the discriminator is replaced with an auto-encoder. The authors hypothesis is that using an auto-encoder helps in 2 ways: 1) denoising expert trajectories for more "robust" learning; 2) using the reconstruction error (instead of binary classification loss) to distinguis experts from samples provides more informative signal for reward learning.

**Strengths**
on a global perspective this work is well motivated
a novel algorithmic variant of GAIL is proposed
thorough experimental evaluation

**weaknesses**
The manuscript doesn't clearly distinguish between adversarial imitation learning algorithms (like GAIL) and "true" inverse reinforcement learning algorithms. This makes it unclear what the real goal of the proposed method is. The ultimate goal of adversarial IL is to learn a policy (by inferring a pseudo-reward at "train" time which is then never used again), while the primary goal of IRL is to learn a reward function at train time, which can then be used at test time. The manuscript motivates the algorithm by saying it will have a more informative signal for learning reward functions, but the algorithm itself is an adversarial IL algorithm which primary goal is to learn a policy from demonstrations. Overall, makes the evaluation and analysis confusing. Ideally, the authors would have focussed on the question "Does the reconstruction error lead to better policies?" (through better pseudo-reward modeling) - or would have extended an IRL method.

Second, the motivation is that the autoencoder helps with more "robust" learning, but it's unclear to me that the evaluation really shows that learning is more robust (also because "robustness" is not clearly defined)

The experimental evaluation is a bit of a mixed bag, and it's a unclear why the new algorithm performs better on non-noisy data (when compared to baselines), but not less so on the noisy data.

**Summary**
Overall, this work provides a promising direction, however in it's current form the manuscript is not yet ready for publication.